# GRAPHGAN:
# GENERATING GRAPHS VIA RANDOM WALKS

## ABSTRACT

We propose GraphGAN – the first implicit generative model for graphs that enables to mimic real-world networks. We pose the problem of graph generation as learning the distribution of biased random walks over a single input graph. Our model is based on a stochastic neural network that generates discrete output samples, and is trained using the Wasserstein GAN objective. GraphGAN enables us to generate *sibling graphs*, which have similar properties yet are not exact replicas of the original graph. Moreover, GraphGAN learns a semantic mapping from the latent input space to the generated graph's properties. We discover that sampling from certain regions of the latent space leads to varying properties of the output graphs, with smooth transitions between them. Strong generalization properties of GraphGAN are highlighted by its competitive performance in link prediction as well as promising results on node classification, even though not specifically trained for these tasks.

## 1 INTRODUCTION

Generative models for graphs have a longstanding history, with applications including data augmentation, anomaly detection and recommendation (Chakrabarti & Faloutsos, 2006). *Explicit probabilistic models* such as Barabási-Albert or stochastic blockmodels are the de-facto standard in this field (Goldenberg et al., 2010). However, it has also been shown on multiple occasions that our intuitions about structure and behavior of graphs may be misleading. For instance, heavy-tailed degree distributions in real graphs were in stark disagreement with the models existing at the time of their discovery (Barabási & Albert, 1999). More recent works, like Dong et al. (2017), keep bringing up other surprising characteristics of real-world networks, not accounted for by the models at hand. This leads us to the question: "How do we define a model that captures all the essential (potentially still unknown) properties of real graphs?"

An increasingly popular way to address this issue in other fields is by switching from *explicit* (prescribed) models to *implicit* ones. This transition is especially notable in Computer Vision, where Variational Autoencoder (Kingma & Welling, 2013) and Generative Adversarial Networks (GANs) (Goodfellow et al., 2014) significantly advanced the state of the art over the classic prescribed approaches like Mixtures of Gaussians (Blanken et al., 2007). GANs achieve unparalleled results in scenarios such as image and 3D objects generation (e.g., Radford et al., 2015; Berthelot et al., 2017; Wu et al., 2016). However, despite their massive success when dealing with real-valued data, adapting GANs to handle *discrete* objects like graphs or text remains an open research problem (Goodfellow, 2016). Indeed, the combinatorial structure of the graph is only one of the obstacles when applying GANs to graphs. Second, large repositories of graphs, which all come from the same distribution, do not exist. This means that in a typical setting one has to learn from a *single graph*. And last, any model operating on a graph necessarily has to be *permutation invariant*, as the graphs remain isomorphic under node reordering.

In this work we introduce *GraphGAN* – the first implicit generative model for graphs, that tackles all of the above challenges. We formulate the problem of learning the graph topology as learning the distribution of biased random walks over the graph. Like in the typical GAN setting, the generator $G$ – in our case defined as a stochastic neural network with discrete output samples – learns to generate random walks that are *plausible* in the real graph, while the discriminator $D$ then has to distinguish them from the true ones that are sampled from the original graph. The objective function of our

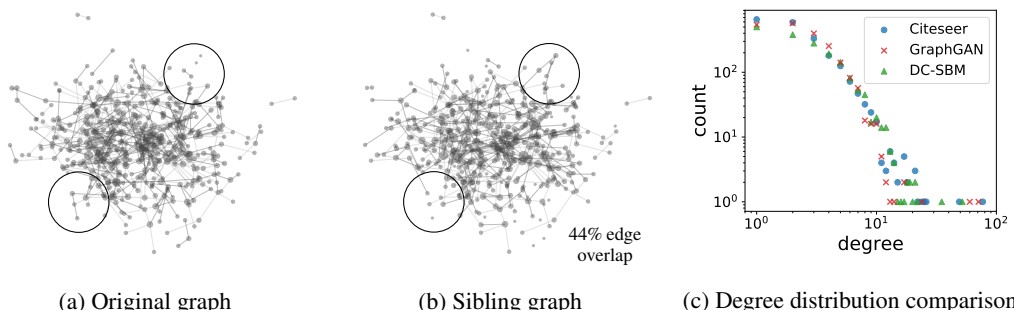

(a) Original graph      (b) Sibling graph      (c) Degree distribution comparison

Figure 1: GraphGAN is able to generate a *sibling graph* to the CITESEER network. (a) and (b) visualize a subset of nodes selected from the complete graphs. The graphs show similar structure but are not identical. (c) shows that the degree distributions of the input graph as well as the graphs generated by GraphGAN and the degree-corrected stochastic blockmodel are very similar.

model is based on the Wasserstein GAN (Arjovsky et al., 2017), which allows to learn multimodal distributions and leads to more stable convergence. Our GraphGAN exhibits strong generalization properties, which we study in detail in the experimental section. The example in Fig. 1 shows that the graphs generated by GraphGAN possess similar properties as the input graph, as shown by the degree distributions in Fig. 1c. The generated graphs, however, are not simply exact replicas: as the visualized subset of nodes in Figs. 1a and 1b shows, the graphs exhibit similar structure while being not identical; in fact, the two graphs have less than 50% of edges in common. This initial insight is underlined by an extensive comparison of graphs generated by GraphGAN and the respective input networks in the experimental section of this work. And even more, when generating graphs based on specific regions of the latent space learned by GraphGAN, we can smoothly interpolate between graphs with varying properties. Our main contributions are:

- We introduce *GraphGAN* - the first of its kind GAN that generates graphs via random walks. Our model tackles the associated challenges of staying permutation invariant, learning from a single graph and generating discrete output.

- We show that our model generalizes, and is able to produce *sibling graphs* to the given input graph. These graphs posses similar topological characteristics, but are not exact replicas (see Fig. 1). We further demonstrate how latent space interpolation leads to generation of graphs with smoothly changing properties.

- We highlight the generalization properties of GraphGAN by its *link prediction* performance, which is competitive with the state of the art on real-word datasets, although not trained explicitly for this task. Additionally, to give the reader a better insight about the behavior of our model, we analyze the learned weights of our model – that can be viewed as node "embeddings" – and by using them in a node classification task we show that they capture meaningful structural information.

## 2 RELATED WORK

There are only two attempts of using GANs in the context of graphs (Tavakoli et al., 2017; Liu et al., 2017). Tavakoli et al. (2017)'s approach tries to generate the full adjacency matrix of the graph directly (treating it as a binary image). To circumvent the issue of having a single graph, they apply random permutations to the adjacency matrix to generate additional training data. Since their model explicitly generates the full adjacency matrix – including zero-elements – the output (input) size of their generator (discriminator) is equal to the number of nodes squared. Such a quadratic complexity is infeasible in practice, allowing to process only small graphs. Indeed, this fundamental limitation becomes evident in their reported runtime of over 60 hours for a graph with only 154 nodes – the largest graph they processed. In contrast, our model operates on the random walks, thus only considering the non-zero elements of the adjacency matrix and efficiently exploiting the sparsity of real-wolrd graphs.

While not focusing on generating full graphs, Liu et al. (2017) use GANs to learn graph topological features. They decompose the graph into multiple subgraphs, where each subgraph is then processed by a GAN using standard image operations (e.g. convolution, deconvolution) which contain the built-in assumption that pixels located closely (within the same receptive field) are in some way correlated. Clearly, when talking about an adjacency matrix such assumption is not sensible, as permutations of rows/columns correspond to the exact same graph but very different receptive fields. In contrast to their approach our model does not make *any* spatial dependence assumptions about the adjacency matrix. Moreover, since their GAN only generates edges *within* the subgraphs, but not *between* them, all inter-subgraph edges from the original graph have to be stored and copied manually to a potential new graph. In contrast, our model is learned end-to-end, it allows to generate the whole graph, and it handles arbitrary permutations of the given input graph.

Due to the challenging nature of the problem, only few approaches able to generate discrete data using GANs exist. Most approaches focus on generating discrete sequences such as text, with some of them using reinforcement learning techniques to address the difficulty of backpropagation through sampling discrete random variables (Yu et al., 2017; Kusner & Hernández-Lobato, 2016; Li et al., 2017; Liang et al., 2017). Other approaches modify the GAN objective to tackle the same challenge (Che et al., 2017; Hjelm et al., 2017). Focusing on non-sequential discrete data, Choi et al. (2017) generate high-dimensional discrete features (e.g. binary indicators, counts) in patient records. None of these methods has considered graph structured data.

Apart from GANs, prescribed generative models for graphs have a long history and are well-studied. For a comprehensive survey see Chakrabarti & Faloutsos (2006); Goldenberg et al. (2010). Based on the different modeling assumptions, the generative power of these models varies significantly. A substantial portion cannot even handle power-law degree distributions as found in most real-world networks. One of the simplest model is the configuration model (Bender & Canfield, 1978; Molloy & Reed, 1995) that rewires edges at random, but preserves the node degrees. A stronger representative able to capture power-law degree distributions, as well as diverse network topologies and community structure is the well-established degree-corrected stochastic blockmodel (DC-SBM) (Karrer & Newman, 2011). Another broad family of models are exponential random graph models (ERGM) (Holland & Leinhardt, 1981) that explicitly preserve some manually specified graph statistics (e.g. edge count, degrees, node attribute statistics, etc.). ERGMs represent a probability distribution over all possible networks of a given fixed size, by specifying one parameter per statistic.

We compare against the configuration model, DC-SBM and ERGM as baselines. We will see in Sec. 4.1 that as expected, all properties which these approaches explicitly model are preserved, while the rest deviate significantly from the input graph. This highlights the need for implicit models such as ours, that capture the properties of real-world graphs without having to manually specify them.

## 3   GANS FOR GRAPHS

In this section we introduce *GraphGAN* - a Generative Adversarial Network model for graphs. Its core idea lies in learning the topology of a graph by learning the distribution over the random walks. Given is an input graph of $N$ nodes, defined by an unweighted adjacency matrix $\boldsymbol{A} \in \{0, 1\}^{N \times N}$. First, we sample a set of random walks of length $T$ from $\boldsymbol{A}$. This collection of random walks serves as a training set for our model. We use the biased second-order random walk sampling strategy described in Grover & Leskovec (2016), as it better captures both local and global graph structure. An important advantage of using random walks is their invariance under node reordering. Additionally, random walks only include the nonzero entries of $\boldsymbol{A}$, thus efficiently exploiting the sparsity of real-world graphs.

Like any typical GAN architecture, GraphGAN consists of two main components - a generator $G$ and a discriminator $D$. The goal of the generator is to generate synthetic random walks that are *plausible* in the input graph. At the same time, the discriminator learns to distinguish the synthetic random walks from the real ones that come from the training set. Both $G$ and $D$ are trained end-to-end using backpropagation. At any point of the training process it is possible to use $G$ to generate a set of random walks, which can then be used to produce an adjacency matrix of a new generated graph. In the rest of this section we describe each stage of this process and our design choices in more detail. An overview of our model's complete architecture can be seen in Fig. 2.

### 3.1 ARCHITECTURE

**Generator.** The generator $G$ defines an implicit probabilistic model for generating the random walks: $(\boldsymbol{v}_1, ..., \boldsymbol{v}_T) \sim G$. We model $G$ as a sequential process based on a neural network $f_\theta$ parametrized by $\theta$. At each step $t$, $f_\theta$ produces two values: the probability distribution over the next node to be sampled, denoted as $\boldsymbol{p}_t$, and the current *memory state* of the model, denoted as $\boldsymbol{m}_t$. The new node $\boldsymbol{v}_t$ (represented as a

$$
\begin{aligned}
& \boldsymbol{z} \sim \mathcal{N}(\boldsymbol{0}, \boldsymbol{I}_d) \\
& \boldsymbol{m}_0 = g_{\theta'}(\boldsymbol{z}) \\
\boldsymbol{v}_1 \sim \text{Cat}(\boldsymbol{p}_1), \quad & (\boldsymbol{p}_1, \boldsymbol{m}_1) = f_\theta(\boldsymbol{m}_0, \boldsymbol{0}) \\
\boldsymbol{v}_2 \sim \text{Cat}(\boldsymbol{p}_2), \quad & (\boldsymbol{p}_2, \boldsymbol{m}_2) = f_\theta(\boldsymbol{m}_1, \boldsymbol{v}_1) \\
\vdots \qquad\quad & \qquad\qquad\qquad \vdots \\
\boldsymbol{v}_T \sim \text{Cat}(\boldsymbol{p}_T), \quad & (\boldsymbol{p}_T, \boldsymbol{m}_T) = f_\theta(\boldsymbol{m}_{T-1}, \boldsymbol{v}_{T-1})
\end{aligned}
$$

one-hot vector) is sampled from $\text{Cat}(\boldsymbol{p}_t)$, and together with $\boldsymbol{m}_t$ passed into $f_\theta$ at the next step $t + 1$. Similarly to the classic GAN setting, a latent code $\boldsymbol{z}$ drawn from a multivariate standard normal distribution is passed through a parametric function $g_{\theta'}$ to initialize $\boldsymbol{m}_0$. The generative process of $G$ is summarized in the box above.

In this work we focus our attention on the Long short-term memory (LSTM) architecture for $f_\theta$, introduced by Hochreiter & Schmidhuber (1997). The memory state $\boldsymbol{m}_t$ of an LSTM is represented by the cell state $\boldsymbol{C}_t$, and the hidden state $\boldsymbol{h}_t$. The latent code $\boldsymbol{z}$ goes through two separate streams, each consisting of two fully connected layers with $\tanh$ activation, and then used to initialize $(\boldsymbol{C}_0, \boldsymbol{h}_0)$.

A natural question might arise: "Why use a model with memory and temporal dependencies, when the random walks are Markov processes?" (2nd order Markov for biased RWs). Or put differently, what's the benefit of using random walks of length greater than 2. In theory, a model with large enough capacity could simply memorize all existing edges in the graph and recreate them. However, for large graphs achieving this in practice is not feasible. More importantly, pure memorization is not the goal of GraphGAN, rather we want to have generalization and generate similar sibling graphs, not exact replicas. Having longer random walks combined with memory helps the model to learn the topology and general patterns in the data (e.g. community structure). Our experiments in Sec. 4.2 confirm this, showing that longer random walks are indeed beneficial.

After each time step, in order to generate the next node in the random walk, the network $f_\theta$ needs to output the vector of probabilities $\boldsymbol{p}_t$ of length $N$. Operating in such high dimensional space within the LSTM cell is infeasible, and leads to unnecessary computational overhead. For this reason, we do the following: the model outputs $\boldsymbol{o}_t \in \mathbb{R}^H$, with $H \ll N$, which is then up-projected to $\mathbb{R}^N$ using $\boldsymbol{W}_{up} \in \mathbb{R}^{N \times H}$. One can view it as similar to context embeddings in representation learning.

Given the probability distribution over the next node in the random walk, $\boldsymbol{p}_t \in \Delta^{N-1}$, from which $\boldsymbol{v}_t$ is to be drawn, we are faced with another challenge: Sampling from a categorical distribution is a non-differentiable operation – thus, it blocks the flow of gradients and precludes backpropagation. We circumvent this problem by using the Straight-Through Gumbel estimator by Jang et al. (2016). More specifically, we perform the following transformation: First, we let

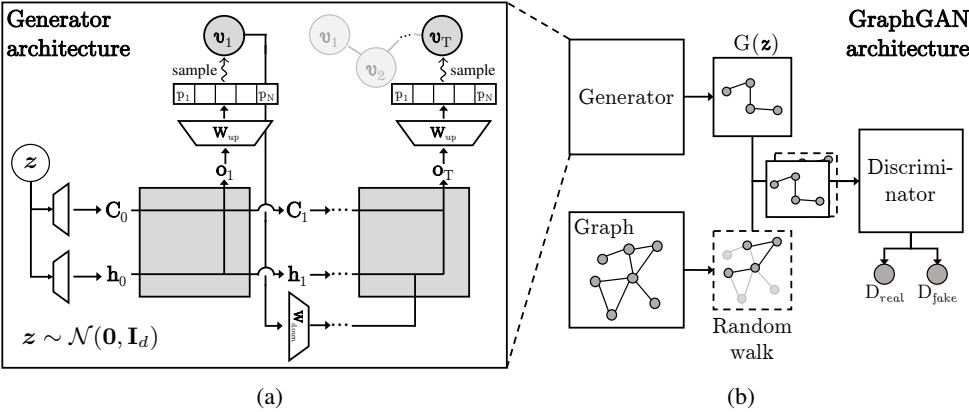

Figure 2: The GraphGAN architecture proposed in this work (b) and the generator architecture (a).

$\boldsymbol{v}_t^* = \text{softmax}\left((\log \boldsymbol{p}_t + \boldsymbol{g})/\tau)\right)$, where $\tau$ is a temperature parameter, and $g_i$'s are i.i.d. samples from a Gumbel distribution with zero mean and unit scale. Then, the next sample is computed as $\boldsymbol{v}_t = \text{onehot}(\arg\max \boldsymbol{v}_t^*)$. While the one-hot sample $\boldsymbol{v}_t$ is passed as input to the next time step, during the backward pass the gradients will flow through the differentiable $\boldsymbol{v}_t^*$. The choice of $\tau$ allows to trade-off between better flow of gradients (large $\tau$, more uniform $\boldsymbol{v}_t^*$) and more exact calculations (small $\tau$, $\boldsymbol{v}_t^* \approx \boldsymbol{v}_t$).

Now that a new node $\boldsymbol{v}_t$ is sampled, it needs to be projected back to a lower-dimensional representation before feeding into the LSTM. This is done by means of down-projection matrix $\boldsymbol{W}_{down} \in \mathbb{R}^{H \times N}$. Together with the up-projection matrix $\boldsymbol{W}_{up}$, the matrix $\boldsymbol{W}_{down}$ deserves a special mention, as it happens to learn – as a byproduct – meaningful information about the nodes. We can view these matrices as a form of node "embeddings" and to gain a better understanding of our model we study their properties in Sec. 4.2 and 4.3.

**Discriminator.** The discriminator $D$ is based on the standard LSTM architecture. At every time step $t$, a one-hot vector $\boldsymbol{v}_t$, denoting the node at the current position, is fed as input. After processing the entire sequence, the discriminator outputs a single score that represents the probability of the random walk being real.

## 3.2 TRAINING

**Wasserstein GAN.** We train our model based on the Wasserstein GAN (WGAN) framework (Arjovsky et al., 2017). To enforce the Lipschitz constraint of the discriminator, we use the gradient penalty as in Gulrajani et al. (2017). We observe that in our setting using the WGAN objective leads to noticeable improvements over the vanilla GAN: it prevents mode collapse, as well as leads to a more stable learning procedure overall. The model parameters $\{\theta, \theta'\}$ are trained using stochastic gradient descent with Adam (Kingma & Ba, 2014). Weights are regularized with an $L_2$ penalty.

**Early stopping.** Because we are interested in generalizing the input graph, the "trivial" solution where the generator has memorized all existing edges is of no interest to us. This means that we need to control overfitting of our model. To achieve this, we employ two early stopping strategies. The first strategy, named VAL-CRITERION is concerned with the generalization properties of GraphGAN. During training, we keep a sliding window of the random walks generated in the last 1,000 iterations and use them to construct a matrix of transition counts. This matrix is then used to evaluate the link prediction performance on a validation set (i.e. ROC and AP scores, for more details see Sec. 4.2). We stop with training when the validation performance stops improving.

The second strategy, named EO-CRITERION makes GraphGAN very flexible and gives the user control over the graph generation. We stop training when we achieve a user specified edge overlap between the generated graphs (see next section) and the original one at a given iteration. Based on her end task the user can choose to generate graphs with either small or large edge overlap with the original, while maintaining structural similarity. This will lead to generated graphs that either generalize better or are closer replicas respectively, yet still capture the properties of the original.

## 3.3 ASSEMBLING THE ADJACENCY MATRIX

After finishing the training, we use the generator $G$ to construct a score matrix $\boldsymbol{S}$ of transition counts, i.e. we count how often an edge appears in the set of generated random walks (typically, using a much larger number of random walks than for early stopping, e.g., 500K). While the raw counts matrix $\boldsymbol{S}$ is sufficient for link prediction purposes, we need to convert it to a binary adjacency matrix $\tilde{\boldsymbol{A}}$, if we wish to reason about the synthetic graph. First, $\boldsymbol{S}$ is symmetrized by setting $s_{ij} = s_{ji} = \max\{s_{ij}, s_{ji}\}$. Because we cannot explicitly control the starting node of the random walks generated by $G$, some high-degree nodes will likely be overrepresented. Thus, a simple binarization strategy like thresholding or choosing top-$k$ entries might lead to leaving out the low-degree nodes and producing singletons. To address this issue, we use the following approach. (i) We ensure that every node $i$ has at least one edge by sampling a neighbor $j$ with probability $p_{ij} = \frac{s_{ij}}{\sum_v s_{iv}}$. If an edge was already sampled before, we repeat the procedure. (ii) We continue sampling edges without replacement, using for each edge $(i, j)$ the probability $p_{ij} = \frac{s_{ij}}{\sum_{u,v} s_{uv}}$, until we reach the desired amount of edges (e.g., as many edges as in the original graph). Note that this procedure is not guaranteed to produce a fully connected graph.

## 4 EXPERIMENTS

Besides using GraphGAN to generate graphs, we also evaluate its output and learned representations on other typical graph mining tasks, most prominently link prediction and node classification. We evaluate GraphGAN on these tasks and several real-world datasets and compare it with state-of-the-art methods. Furthermore, we demonstrate how we can generate graphs with smoothly changing properties via latent space interpolation.

**Datasets.** For our evaluation, we use several well-know citation datasets, as well as the Political Blogs dataset. Table 1 show the dataset statistics. Cora-ML is the subset of machine learning papers from the original Cora dataset typically considered in other works. For all our experiments, we consider only the largest connected component in each network and treat them as undirected.

Table 1: Dataset statistics

|  | Pol. Blogs | Cora-ML (McCallum et al., 2000) | Cora (Giles et al., 1998) | Citeseer (Sen et al., 2008) | Pubmed (Pan et al., 2016) | DBLP |
|---|---|---|---|---|---|---|
| Number of nodes | 1,490 | 2,995 | 19,793 | 3,312 | 19,717 | 17,716 |
| Largest conn. comp. size | 1,222 | 2,810 | 18,800 | 2,110 | 19,717 | 16,191 |
| Number of edges | 19,025 | 8,416 | 65,311 | 4,715 | 44,324 | 52,867 |
| Edges in LCC | 16,714 | 8,229 | 64,529 | 3,757 | 44,324 | 51,913 |
| Number of communities | 2 | 7 | 70 | 6 | 3 | 4 |

### 4.1 GRAPH GENERATION

In this task, we use GraphGAN to generate *sibling graphs* to a given input graph, and compare its performance to the baselines. The goal is to generate graphs that are similar in their properties to the input graph – while not trivially copying the input network. We randomly hide $15\%$ of the edges (which are used for the stopping criterion; see Sec. 3.2) and train GraphGAN and DC-SBM on the remaining graph. We then sample graphs from the trained models and compare their properties with the input graph. We report the results for CORA-ML here, and for CITESEER in the appendix.

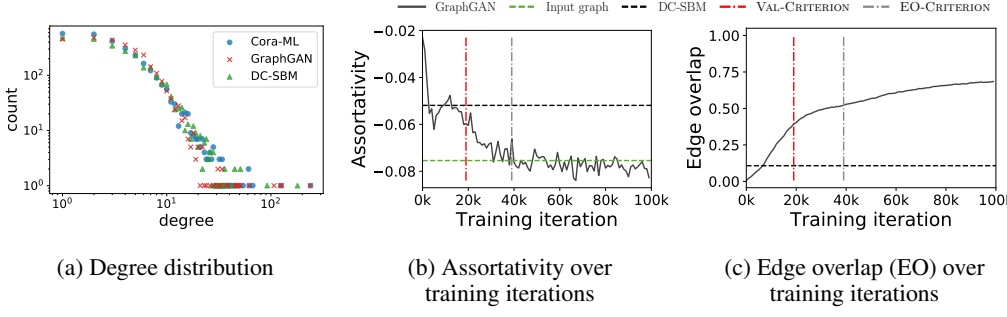

(a) Degree distribution

(b) Assortativity over training iterations

(c) Edge overlap (EO) over training iterations

Figure 3: Properties of generated graphs, trained on CORA-ML.

**Evaluation.** Fig. 3a shows that GraphGAN and DC-SBM are able to generate graphs whose degree distributions nicely match the input graph's. For DC-SBM, this is not surprising, given that it *explicitly* encodes the degree distribution in the model. GraphGAN, however, does not have access to the degree distribution of the input graph – yet is still able to model it to a high accuracy. Going beyond the degree distribution, in Table 2 we show seven other important graph statistics and see that for most of them GraphGAN closely matches the original graph. We do not expect that GraphGAN is superior to every existing *explicit* model in every possible regard. Rather, our goal is to lay a foundation for the study of *implicit* models for graph generation. We report the results for both early stopping strategies: VAL-CRITERION and EO-CRITERION. In line with our intuition, we can see that higher EO leads to generated graphs with statistics closer to the original. Results for additional statistics, respective definitions, as well as details about the baselines can be found in the appendix.

Figs. 3b and 3c show how the graph statistics evolve as we train GraphGAN on CORA-ML. In Fig. 3b we see that after 40K training iterations we are able to reach the assortativity value of the original graph. Fig. 3c shows that the edge overlap smoothly increasing with the number of epochs.

Table 2: Comparison of graph statistics between the original CORA-ML graph and graphs generated by GraphGAN and the baselines, averaged over 5 trials. Marked in bold and italics are the results that are closest and second-closest to the input graph, respectively. * indicates values for the configuration model that by definition exactly match the input graph.

| Graph | Max. degree | | Assorta-tivity | | Triangle count | | Power law exponent | | Largest conn. comp. | | Avg. Inter-community density | | Avg. Intra-community density | |
|---|---|---|---|---|---|---|---|---|---|---|---|---|---|---|
| | Avg. | Std. | Avg. | Std. | Avg. | Std. | Avg. | Std. | Avg. | Std. | Avg. | Std. | Avg. | Std. |
| CORA-ML | 240 | | -0.075 | | 2,814 | | 1.86 | | 2,810 | | $4.3e{-}4$ | | $1.7e{-}3$ | |
| Conf. model (1% EO) | * | * | -0.030 ± 0.003 | | 322 | ± 31 | * | * | 2,785 ± 4.9 | | $1.6e{-}3$ | ±1e−5 | $2.8e{-}4$ | ±1e−5 |
| Conf. model (52% EO) | * | * | -0.051 ± 0.002 | | 626 | ± 19 | * | * | 2793 | ± 6.0 | $9.8e{-}4$ | ±1e−5 | $9.9e{-}4$ | ±2e−5 |
| node2vec naïve (1% EO) | 14 | ± 1.4 | -0.007 ± 0.011 | | 16 | ± 4.4 | 1.68 | ± 0.001 | **2,810** ± 0.1 | | $1.4e{-}3$ | ±1e−5 | $3.8e{-}4$ | ±2e−5 |
| DC-SBM (11% EO) | 165 | ± 9.0 | -0.052 ± 0.004 | | 1,403 | ± 67 | **1.814** ± 0.008 | | 2,474 ± 18.9 | | $6.7e{-}4$ | ±2e−5 | $1.2e{-}3$ | ±4e−5 |
| ERGM (56% EO) | **243** | ± 1.94 | **-0.077** ± 0.000 | | **2,293** ± 23 | | 1.786 ± 0.003 | | 2,489 ± 11.0 | | $6.9e{-}4$ | ±2e−5 | $1.2e{-}3$ | ±1e−5 |
| GraphGAN VAL (39% EO) | 199 | ± 6.7 | -0.060 ± 0.004 | | 1,410 ± 30 | | 1.773 ± 0.002 | | **2,809** ± 1.6 | | *6.5e−4* | ±1e−5 | *1.3e−3* | ±2e−5 |
| GraphGAN EO (52% EO) | *233* | ± 3.6 | *-0.066* ± 0.003 | | *1,588* ± 59 | | *1.793* ± 0.003 | | *2,807* ± 1.6 | | **6.0e−4** | ±1e−5 | **1.4e−3** | ±1e−5 |

Note: EO is a suitable measure of closeness since we generate graphs with same node ordering and same number of edges as the input. We provide similar plots for the other graph statistics and for CITESEER in the appendix.

## 4.2 LINK PREDICTION

Link prediction is a classical task in graph mining, where the goal is to predict new links in a given graph. We use it to evaluate the generalization properties of GraphGAN. We hold out 10% of edges from the graph for validation, and 5% as the test set, along with the same amount of randomly selected non-edges. We also ensure that the training network remains connected and does not contain any singletons. We measure the performance with the commonly used metrics area under the ROC curve (AUC) score and precision-recall AUC score, known as average precision (AP).

To evaluate GraphGAN's link prediction performance, we sample a specific number of random walks (500K/100M) from the trained generator. We use the observed transition counts between any two nodes as a measure of how likely there is an edge between them. Additionally, we concatenate $W_{down}$ and $W_{up}$ and use the dot product as an alternative way to perform link prediction. We compare with Adamic/Adar (Adamic & Adar, 2003), the degree-corrected stochastic blockmodel (DC-SBM) (Karrer & Newman, 2011), and node2vec (Grover & Leskovec, 2016).

**Evaluation.** The results are listed in Table 3. There is no overall dominating method, with different methods achieving best results on different datasets. GraphGAN shows competitive performance for all datasets, for both the transition count based and the dot product based link prediction, even achieving state-of-the-art results for some of them, despite not being explicitly trained for this task.

Table 3: Link prediction performance.

| Method | CORA-ML | | CORA | | CITESEER | | DBLP | | PUBMED | | POLBLOGS | |
|---|---|---|---|---|---|---|---|---|---|---|---|---|
| | ROC | AP | ROC | AP | ROC | AP | ROC | AP | ROC | AP | ROC | AP |
| Adamic/Adar | 92.16 | 85.43 | 93.00 | 86.18 | 88.69 | 77.82 | 91.13 | 82.48 | 84.98 | 70.14 | 85.43 | 92.16 |
| DC-SBM | **96.03** | 95.15 | 98.01 | 97.45 | 94.77 | 93.13 | **97.05** | **96.57** | **96.76** | 95.64 | 95.46 | **94.93** |
| node2vec | 92.19 | 91.76 | **98.52** | **98.36** | 95.29 | 94.58 | 96.41 | 96.36 | 96.49 | **95.97** | 85.10 | 83.54 |
| GraphGAN (500K) | 94.00 | 92.32 | 82.31 | 68.47 | 95.18 | 91.93 | 82.45 | 70.28 | 87.39 | 76.55 | 95.06 | 94.61 |
| GraphGAN (100M) | 95.19 | **95.24** | 84.82 | 88.04 | **96.30** | **96.89** | 86.61 | 89.21 | 93.41 | 94.59 | **95.51** | 94.83 |
| GraphGAN ($[W_{down}, W_{up}]$) | 90.29 | 88.29 | 84.38 | 79.36 | 92.95 | 92.44 | 86.59 | 81.96 | 91.79 | 89.37 | 70.01 | 62.72 |

Interestingly, for the transition count based link prediction, the GraphGAN performance increases when increasing the number of random walks sampled from the generator. This is especially true for the larger networks (CORA, DBLP, PUBMED), since given their size we need more random walks to cover the entire graph. This suggests that for an additional computational cost, one can get significant gains in performance. Note that while 100M may seem like an large number, the sampling process is trivially parallelizable.

**Sensitivity analysis.** Although GraphGAN has many hyperparameters – typical for a GAN model, in practice most of them are not critical for performance. The two important exceptions are the length of the random walks $T$, and the discriminator type. Fig. 4 empirically confirms the choice of a neural network that generates random walks of length $T$ as opposed to just edges; the model does not have

the capacity to fit the model by just considering edges (i.e. random walks of length 2). In the experiment we sample 500K random walks from the trained models for each $T$, averaged over five runs. The performance gain for random walk length 20 over 16 is marginal and does not outweigh the additional computational cost; therefore, we use random walks of length 16 for all experiments. In the appendix we show that our choice of a recurrent discriminator achieves better link prediction performance than a variant based on convolutions, hence we use it for all experiments in this work.

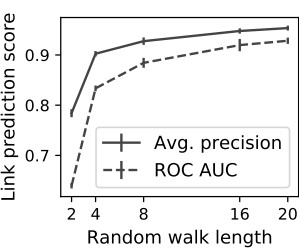

Figure 4: Effect of the random walk length on the performance.

### 4.3 NODE CLASSIFICATION

We perform node classification using $\boldsymbol{W}_{up}$ and $\boldsymbol{W}_{down}$, which can be viewed as low-dimensional feature representations of the nodes, to show the generalization properties of GraphGAN. We evaluate both $\boldsymbol{W}_{down}$, $\boldsymbol{W}_{up}$, as well as their concatenation, comparing with node2vec as a strong baseline. Note that unlike in the link prediction task we cannot use the generated graphs to perform node classification. Similar to (Grover & Leskovec, 2016; Perozzi et al., 2014) we train a logistic regression model on a small randomly selected subset ($< 20\%$ of nodes) using the ground-truth labels and evaluate the classification performance on the remaining samples. Additionally, we visualize $\boldsymbol{W}_{up}$ using t-SNE to demonstrate that a community structure has emerged in the embedding space.

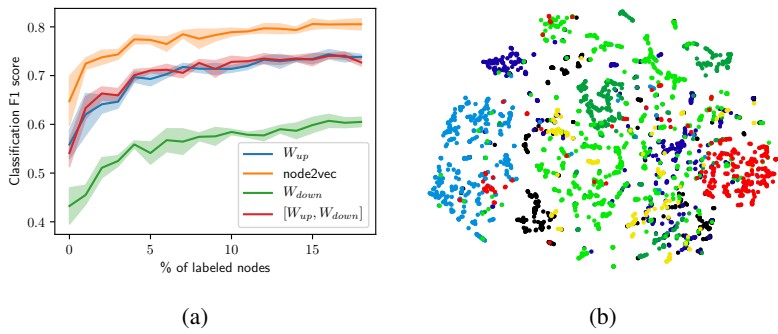

(a)  (b)

Figure 5: (a) Node classification performance on CORA-ML, and (b) t-SNE visualization of $\boldsymbol{W}_{up}$, where the color indicates the ground truth communities.

**Evaluation.** In Fig. 5a, we visualize the weighted macro F1 score for node classification. The results are averaged over five trials, and the shaded areas indicate the standard deviation of the respective curves. The embedding-like weights in $\boldsymbol{W}_{up}$ produced by GraphGAN as a byproduct show comparable performance with node2vec – an algorithm specifically designed to learn node embeddings. This indicates that during the process of learning to generate random walks GraphGAN also captured meaningful structural information. The emergent community structure seen in Fig. 5b further emphasizes this. In this t-SNE visualization of $\boldsymbol{W}_{up}$, we can clearly see a grouping of nodes from the same community, which means that they lie in similar regions of the embedding space. Although GraphGAN's learned node representations clearly show they they capture useful information about the nodes, they are only a byproduct of the training procedure. Recall that the main goal of GraphGAN is graph generation; if node classification is the user goal, dedicated node embedding algorithms are the preferred option.

### 4.4 LATENT VARIABLE INTERPOLATION

Latent space interpolation is a good way to gain insight into what kind of structure the generator was able to capture. To be able to visualize the properties of the generated graphs we train our model using noise $\boldsymbol{z}$ drawn from a bivariate standard normal distribution, which corresponds to a 2-dimensional latent space $\Omega = \mathbb{R}^2$. Then, instead of sampling $\boldsymbol{z}$ from the entire latent space $\Omega$, we now sample from subregions of $\Omega$ and visualize the results. More specifically, we divide $\Omega$ into $20 \times 20$ subregions (bins) of equal probability mass using the cumulative distribution function $\Phi$. For

each bin we generate 62.5K random walks. We evaluate properties of both the generated random walks themselves, as well as properties of the resulting graph when sampling a binary adjacency matrix for each bin, visualizing them as heatmaps.

**Evaluation.** In Fig. 6a and 6b we see properties of the generated random walks; in Fig. 6c and 6d, we visualize properties of graphs sampled from the random walks in the respective bins. In all four heatmaps, we see distinct patterns, e.g. higher average degree of starting nodes for the bottom right region of Fig. 6a, or higher degree distribution inequality in the top-right area of Fig. 6c. While Fig. 6c and 6d show that certain regions of $z$ correspond to generated graphs with very different degree distributions, recall that sampling from the entire latent space ($\Omega$) yields *sibling* graphs with degree distribution similar to the original graph (see Fig. 1c). The model was trained on CORA-ML. We provide further heatmaps for other metrics (16 in total) as well as visualizations for CITESEER in the appendix.

This experiment clearly demonstrates that by interpolating in the latent space we can obtain graphs with smoothly changing properties. The smooth transitions in the heatmaps provide evidence that our model learns to map specific parts of the latent space to specific properties of the graph.

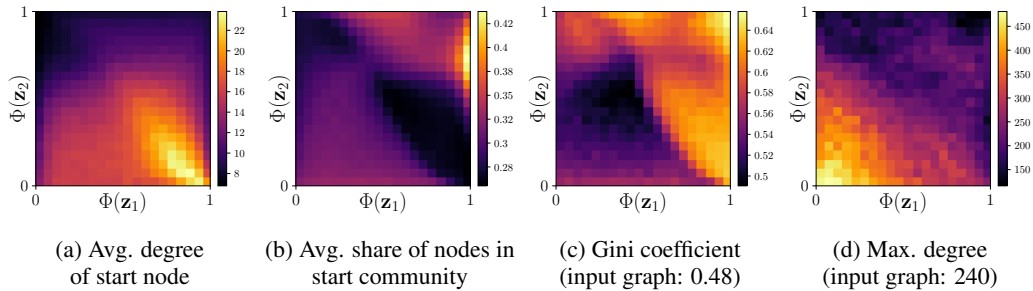

(a) Avg. degree
of start node

(b) Avg. share of nodes in
start community

(c) Gini coefficient
(input graph: 0.48)

(d) Max. degree
(input graph: 240)

Figure 6: Properties of the random walks as well as the graphs sampled from the $20 \times 20$ bins. 6a and 6b show properties of the random walks, 6c and 6d show properties of the generated graphs.

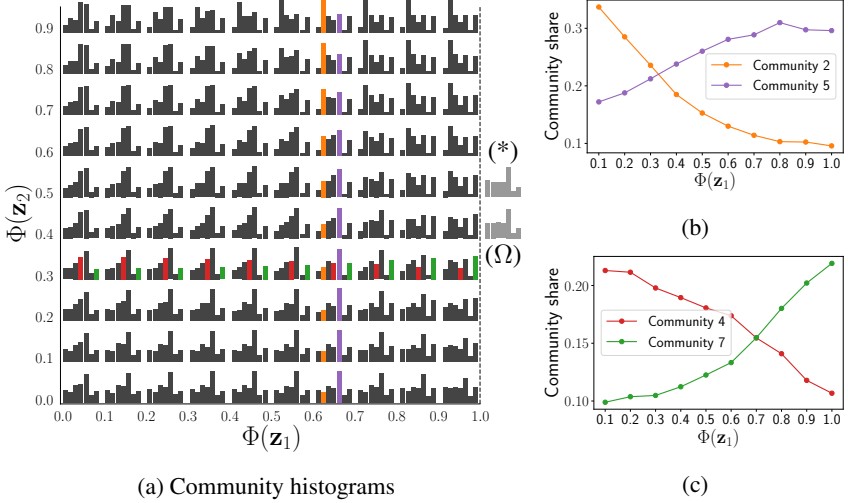

(a) Community histograms

(b)

(c)

Figure 7: Community distributions (see appendix for definition) when sampling random walks on subsets of the latent space $z$. (a) shows complete community histograms on a $10 \times 10$ grid. (b) and (c) show exemplary trajectories in latent space. ($\Omega$) is the community distribution when sampling from the entire latent space, and (*) is the community distribution of the CORA-ML network. Also available as an animation https://goo.gl/BGDX4o.

We can also see this mapping from latent space to the generated graph properties in the community distribution histograms on a $10 \times 10$ grid in Fig. 7. Marked by (*) and ($\Omega$) we see the community distributions for the input graph and the graph obtained by sampling on the complete latent space,

respectively. In Fig. 7b and 7c, we see the evolution of selected community shares when following a trajectory from top to bottom, and left to right, respectively. The community histograms resulting from sampling random walks from opposing regions of the latent space are very different; again the transitions between these histograms are smooth, as can be seen in the trajectories in Fig. 7b and 7c.

## 5 DISCUSSION AND FUTURE WORK

When evaluating different graph generative models in Sec. 4.1, we observed a major limitation of explicit models. While the prescribed approaches excel at recovering the properties that are directly included in their definition, they perform significantly worse with respect to the rest of the metrics. This phenomenon clearly indicates the need for implicit graph generators, such as GraphGAN. Indeed, we notice that our model is able to consistently capture all the important graph characteristics (see Table 2). Moreover, GraphGAN generalizes beyond the input graph, as can be seen by its strong link prediction performance in Sec. 4.2.

Still, being the first model of its kind, GraphGAN possesses certain limitations, and a number of related questions could be addressed in follow-up works:

**Scalability.** We have observed in Sec. 4.2 that it takes a large number of generated random walks to get representative transition counts for large graphs. While sampling random walks from GraphGAN is trivially parallelizable, a possible extension of our model is to use a *conditional* generator, i.e. the generator can be provided a desired starting node, thus ensuring a more even coverage. On the other hand, the sampling procedure itself can be sped up by incorporating a hierarchical softmax output layer - a method commonly used in natural language processing.

**Evaluation.** It is nearly impossible to judge whether a graph is realistic by visually inspecting it (unlike images, for example). In this work we already quantitatively evaluate the performance of GraphGAN on a large number of standard graph statistics. However, developing new measures applicable to (implicit) graph generative models will deepen our understanding of their behavior.

**Experimental scope.** In the current work we focused on the setting of a single connected graph. Other scenarios, such as dealing with a collection of smaller i.i.d. graphs, that frequently occur in other fields (e.g., chemistry, biology), would be an important application area for the proposed model. Studying the influence of the graph topology (e.g., sparsity, diameter) on performance of GraphGAN will shed more light on its properties.

**Other types of graphs.** While plain graphs are ubiquitous, many of real-world applications deal with attributed, k-partite or heterogeneous networks. Adapting the GraphGAN model to handle these other modalities of the data is a promising direction for future research. Especially important would be an adaptation to the dynamic / inductive setting, when new nodes are added over time.

## 6 CONCLUSION

GraphGAN is the first work to successfully bridge the worlds of implicit modeling and graphs. Our work enables future researchers to gain better insight into the properties of real networks and opens new and exciting lines of research. We are able to generate realistic graphs by learning to generate (biased) random walks from the same distribution as the random walks from an input graph. We employ the GAN framework to learn our implicit generative model, overcoming key challenges such as permutation invariance, working in the discrete domain and having a single graph as input. Our generator is able to generate sibling graphs that maintain structural similarity with the original graph without being exact replicas. Better yet, using our defined stopping criteria, we can control how close are the generated graphs to the original. We further show that GraphGAN learns a semantic mapping from the latent space to the properties of the generated graph, which is evidenced by the smooth transitions of the output. GraphGAN shows strong generalization properties, as demonstrated by the competitive performance on the link prediction and the promising results on the node classification task, without being explicitly trained with these tasks in mind.

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

APPENDIX

We denote as $V$ the set of all nodes in a graph, $E$ as the set of edges, and as $C_i \subseteq V$, $i \in \{1, \ldots, k\}$ communities in a graph. Every node in the graphs we consider is assigned to exactly one of these communities. $N(v) = \{v'|(v, v') \in E\}$ denotes the set of neighbors of a node $v$, and $d(v) = |N(v)|$ is the degree of node $v$.

Given two graphs $G_1$ and $G_2$ with the same number of nodes and edges, i.e. $|V_1| = |V_2|$, $|E_1| = |E_2|$, we define their *edge overlap* as

$$\text{EO}(G_1, G_2) = \frac{|E_1 \cap E_2|}{|E_1|} = \frac{|E_1 \cap E_2|}{|E_2|}.$$

GRAPH GENERATION BASELINES

**Graph generation from node embeddings.** As a naïve baseline for generating a graph from node embeddings, we propose the following strategy. As suggested by Grover & Leskovec (2016), we train a logistic regression model on node embeddings learned by node2vec to get edge probabilities. Using these link prediction scores, we generate first-order random walks. The starting nodes are sampled from a categorical distribution where the probability for each node is proportional to its degree in the input graph. The subsequent nodes are sampled using the logistic regression model, i.e. proportional to the log probabilities of a link between the previous node and all other nodes. We repeat this for T=16 time steps and for 500K random walks. We use our procedure described in Section 3.3 to assemble an adjacency matrix from the transition counts.

**Configuration model.** In addition to randomly rewiring *all* edges in the input graph, we also generate random graphs with similar overlap as graphs generated by GraphGAN using the configuration model. For this, we randomly select a share of edges (e.g. 39%) and keep them fixed, and shuffle the remaining edges. This leads to a graph with the specified edge overlap; in Table 2 we show that with the same edge overlap, GraphGAN's generated graphs in general match the input graph better w.r.t the statistics we measure.

**Exponential random graph model.** The ERGM we used takes as parameters the edge count, density, degree correlation, deg1.5, and gwesp. Here, deg1.5 is the sum of all degrees to the power of 1.5, and gwesp refers to the geometrically weighted edgewise shared partner distribution (see Handcock et al. (2017) for details).

Table 4: Graph statistics used to measure graph properties in this work.

| Metric name | Computation | Description |
|---|---|---|
| Maximum degree | $\max\limits_{v \in V} d(v)$ | Maximum degree of all nodes in a graph. |
| Community distribution | $c_i = \frac{\sum_{v \in C_i} d(v)}{\sum_{v \in V} d(v)}$ | Share of in- and outgoing edges of community $C_i$, normalized by the number of edges in the graph. |
| LCC | $N_{max} = \max\limits_{f \subseteq F} |f|$ | Size of largest connected component, where $F$ are all connected components. |
| Power law exponent | $1 + n \left( \sum\limits_{u \in V} \log \frac{d(u)}{d_{\min}} \right)^{-1}$ | Exponent of the power law distribution, where $d_{min}$ denotes the minimum degree in a network. |
| Gini coefficient | $\frac{2 \sum_{i=1}^{|V|} i \hat{d}_i}{|V| \sum_{i=1}^{|V|} \hat{d}_i} - \frac{|V|+1}{|V|}$ | Common measure for inequality in a distribution, where $\hat{d}$ is the sorted list of degrees in the graph. |
| Triangle count | $\frac{|\{\{u,v,w\}|\{(u,v),(v,w),(u,w)\} \subseteq E\}|}{6}$ | Number of triangles in the graph, where $u \sim v$ denotes that $u$ and $v$ are connected. |
| Wedge count | $\sum_{v \in V} \binom{d(v)}{2}$ | Number of wedges, i.e. two-hop paths in an undirected graph. |
| Rel. edge distr. entropy | $\frac{1}{\ln |V|} \sum_{v \in V} -\frac{d(v)}{|E|} \ln \frac{d(v)}{|E|}$ | Entropy of degree distribution, 1 means uniform, 0 means a single node is connected to all others. |
| Assortativity | $\rho = \frac{\text{cov}(X,Y)}{\sigma_X \sigma_Y}$ | Pearson correlation of degrees of connected nodes, where the $(x_i, y_i)$ pairs are the degrees of connected nodes. |

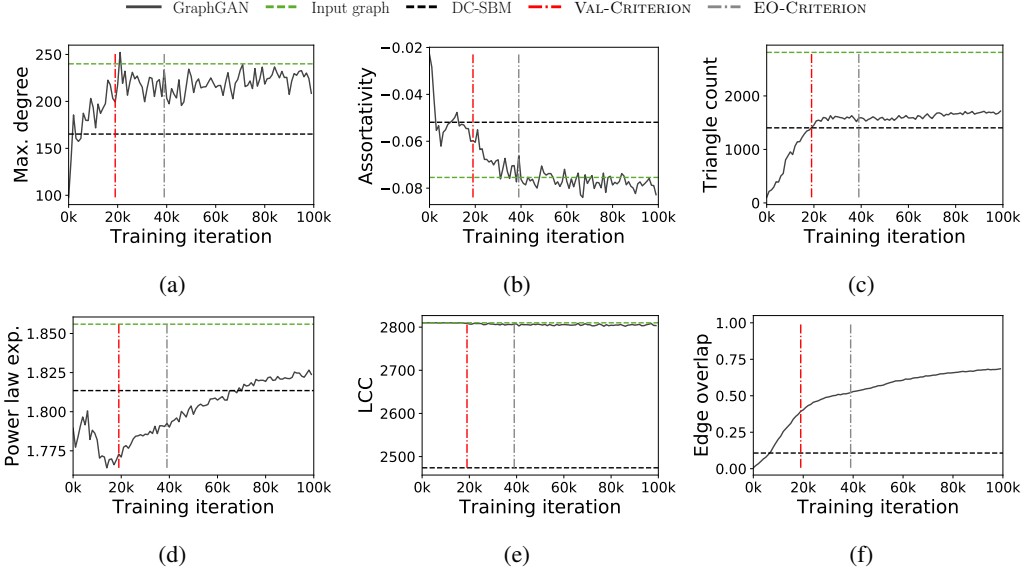

Figure 8: Evolution of graph statistics during training on CORA-ML

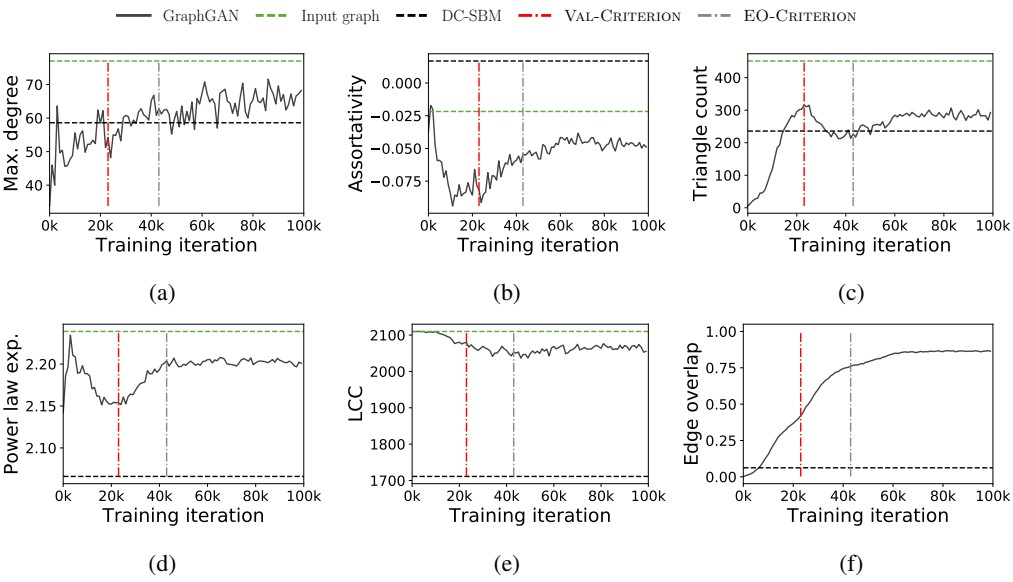

Figure 9: Evolution of graph statistics during training on CITESEER

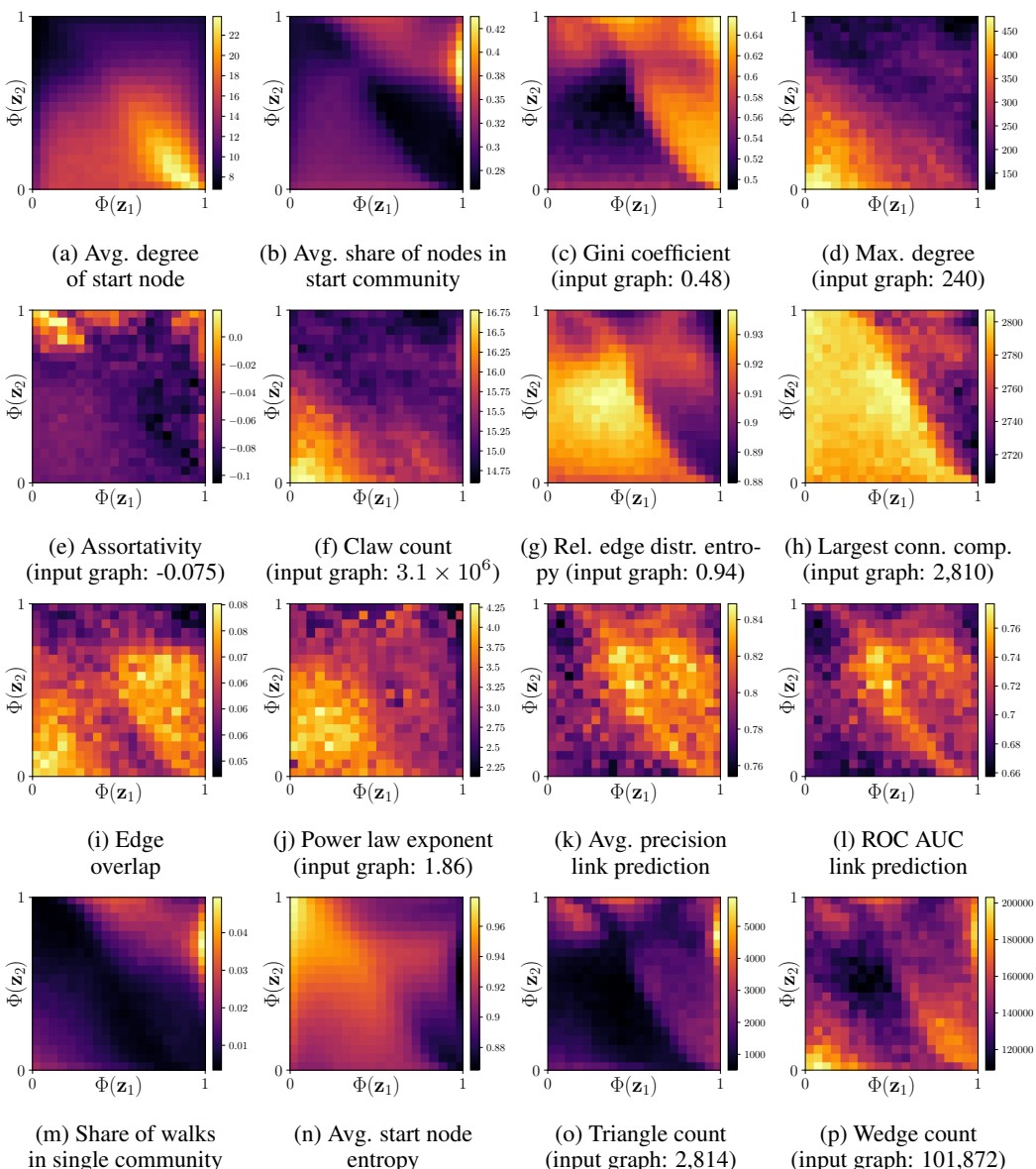

Figure 10: Properties of the random walks as well as the graphs sampled from the $20 \times 20$ latent space bins, trained on CORA-ML.

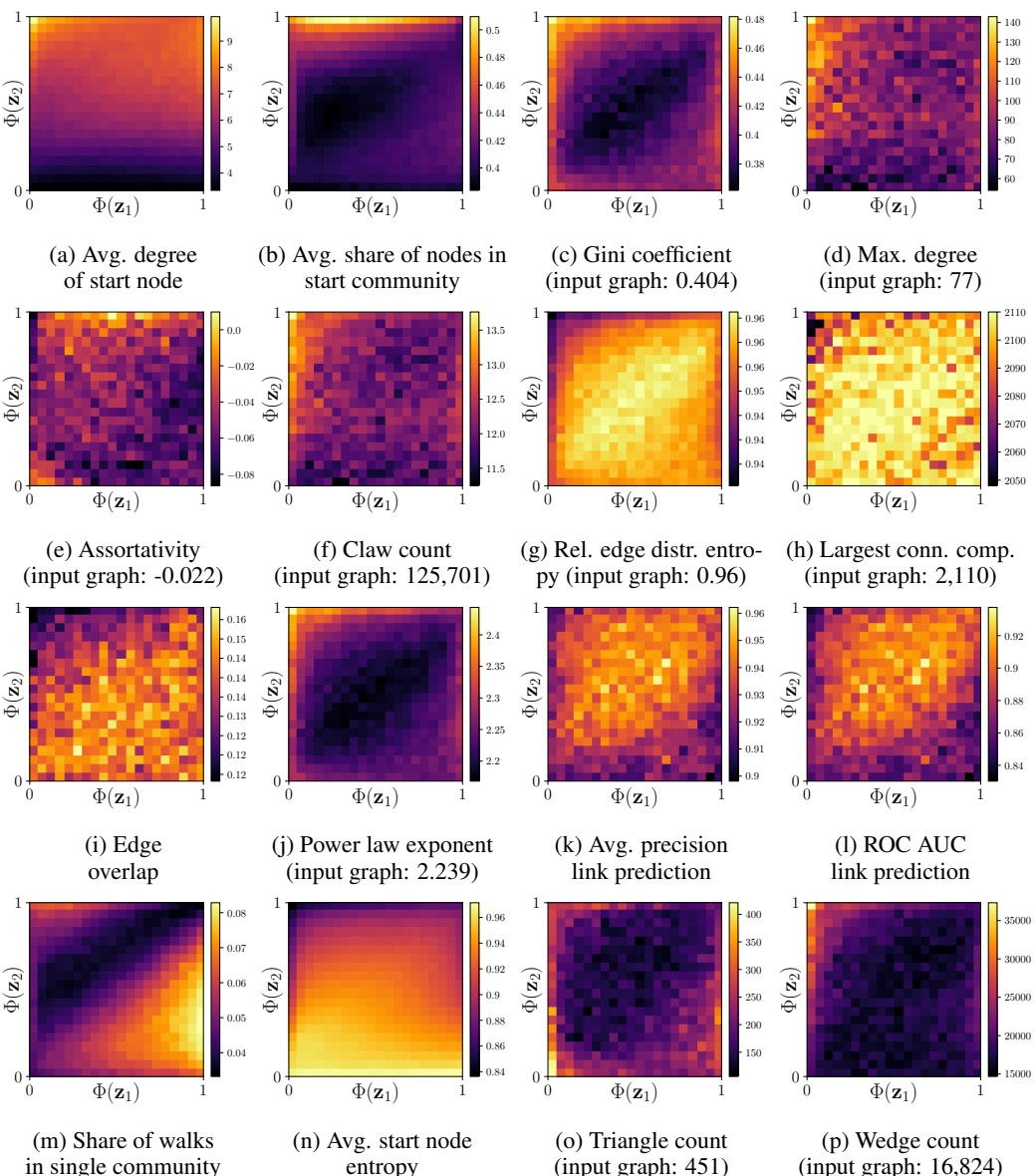

Figure 11: Properties of the random walks as well as the graphs sampled from the $20 \times 20$ latent space bins, trained on CITESEER.

Table 5: GraphGAN with recurrent vs convolutional discriminator. We train GraphGAN with the recurrent and convolutional discriminator variants five times each and measure their link prediction scores on the CORA-ML dataset to evaluate which variant is suited better for our task.

| Discri-minator | ROC AUC Mean Std. | Avg. Prec. Mean Std |
|---|---|---|
| Recurrent | **92.07** ± 0.005 | **94.88** ± 0.002 |
| Conv. | 89.70 ± 0.017 | 93.02 ± 0.011 |

Table 6: Comparison of graph statistics between the CITESEER/CORA-ML graph and graphs generated by GraphGAN and DC-SBM, averaged after 5 trials. Marked in bold and italic are the results that are closest and second-closest to the ground truth graph, respectively, except for edge overlap, where lower can be considered better.

| Graph | Max. degree Avg. | Std. | Assortativity Avg. | Std. | Triangle count Avg. | Std. | Power law exponent Avg. | Std. | Avg. Inter-community density Avg. | Std. | Avg. Intra-community density Avg. | Std. |
|---|---|---|---|---|---|---|---|---|---|---|---|---|
| CITESEER | 77 | | -0.022 | | 451 | | 2.239 | | 4.9e−4 | | 9.3e−4 | |
| GraphGAN (42% EO) | 54 | ± 4.2 | -0.082 ± 0.009 | | *316* | ± 11.2 | *2.154* ± 0.003 | | 6.5e−4 | ±2e−5 | *8.0e−4* | ±2e−5 |
| GraphGAN (76% EO) | *63* | ± 4.3 | **-0.054** ± 0.006 | | 227 | ± 13.3 | **2.204** ± 0.003 | | **5.9e−4** | ±2e−5 | **8.6e−4** | ±1e−5 |
| DC-SBM (6.6% EO) | 53 | ± 5.6 | **0.022** ± 0.018 | | 257 | ± 30.9 | 2.066 ± 0.014 | | 7.6e−4 | ±2e−5 | 5.3e−4 | ±3e−5 |
| Conf. model | * | * | -0.017 ± 0.006 | | 20 | ± 6.50 | * | * | 1.1e−3 | ±1e−5 | 2.3e−4 | ±2e−5 |
| Conf. model (42% EO) | * | * | *-0.020* ± 0.009 | | 54 | ± 8.8 | * | * | 8.4e−4 | ±1e−5 | 5.1e−4 | ± 1e−5 |
| Conf. model (76% EO) | * | * | -0.024 ± 0.006 | | 207 | ± 11.8 | * | * | *6.3e−4* | ±1e−5 | 7.6e−4 | ± 1e−5 |
| node2vec naïve | 9 | ± 0.4 | -0.052 ± 0.021 | | 2 | ± 0.49 | 2.04 ± 0.002 | | 1.1e−3 | ±2e−5 | 2.7e−4 | ±1e−5 |
| ERGM (27% EO) | **66** | ± 1 | 0.052 ± 0.005 | | **415.6** ± 8 | | 2.0 ± 0.01 | | 9.3e−4 | ±2e−5 | 4.8e−4 | ±6e−6 |
| CORA-ML | 240 | | -0.075 | | 2,814 | | 1.86 | | 4.3e−4 | | 1.7e−3 | |
| GraphGAN (39% EO) | 199 | ± 6.7 | -0.060 ± 0.004 | | 1,410 ± 30 | | 1.773 ± 0.002 | | *6.5e−4* | ±1e−5 | *1.3e−3* | ±2e−5 |
| GraphGAN (52% EO) | *233* | ± 3.6 | *-0.066* ± 0.003 | | *1,588* ± 59 | | 1.793 ± 0.003 | | **6.0e−4** | ±1e−5 | **1.4e−3** | ±1e−5 |
| DC-SBM (11% EO) | 165 | ± 9.0 | -0.052 ± 0.004 | | 1,403 ± 67 | | **1.814** ± 0.008 | | 6.7e−4 | ±2e−5 | 1.2e−3 | ±4e−5 |
| Conf. model | * | * | -0.030 ± 0.003 | | 322 | ± 31 | * | * | 1.6e−3 | ±1e−5 | 2.8e−4 | ±1e−5 |
| Conf. model (39% EO) | * | * | -0.050 ± 0.005 | | 420 | ± 14 | * | * | 1.1e−3 | ±1e−5 | 8.0e−4 | ±1e−5 |
| Conf. model (52% EO) | * | * | -0.051 ± 0.002 | | 626 | ± 19 | * | * | 9.8e−4 | ±1e−5 | 9.9e−4 | ±2e−5 |
| node2vec naïve | 14 | ± 1.4 | -0.007 ± 0.011 | | 16 | ± 4.4 | 1.68 ± 0.001 | | 1.4e−3 | ±1e−5 | 3.8e−4 | ±2e−5 |
| ERGM (56% EO) | **243** | ± 1.94 | **-0.077** ± 0.000 | | **2,293** ± 23 | | 1.786 ± 0.003 | | 6.9e−4 | ±2e−5 | 1.2e−3 | ±1e−5 |

| Graph | Wedge count Avg. | Std. | Rel. edge distr. entr. Avg. | Std. | Largest conn. comp Avg. | Std. | Claw count Avg. | Std. | Gini coeff. Avg. | Std. | Edge overlap Avg. | Std. |
|---|---|---|---|---|---|---|---|---|---|---|---|---|
| CITESEER | 16,824 | | 0.959 | | 2,110 | | 125,701 | | 0.404 | | 1 | |
| GraphGAN (42% EO) | 12,998 | ± 84.6 | 0.969 ± 0.000 | | **2,079** ± 12.6 | | 57,654 ± 4,226 | | 0.354 ± 0.001 | | 0.42 | ± 0.006 |
| GraphGAN (76% EO) | *15,202* | ± 378 | 0.963 ± 0.000 | | 2,053 ± 23 | | **94,149** ± 11,926 | | **0.385** ± 0.002 | | 0.76 | ± 0.01 |
| DC-SBM (6.6% EO) | *15,531* | ± 592 | 0.938 ± 0.001 | | 1,697 ± 27 | | 69,818 ± 11,969 | | 0.502 ± 0.005 | | 0.066 ± 0.011 |
| Conf. model | * | * | 0.955 ± 0.001 | | 2,011 ± 6.8 | | * | * | * | * | 0.008 ± 0.001 |
| Conf. model (42% EO) | * | * | *0.956* ± 0.001 | | 2,045 ± 12.5 | | * | * | * | * | 0.42 | ± 0.002 |
| Conf. model (76% EO) | * | * | **0.957** ± 0.001 | | *2,065* ± 10.2 | | * | * | * | * | 0.76 | ± 0.0 |
| node2vec naïve | 8,157 | ± 36.0 | 0.986 ± 0.000 | | 2,110 ± 0.0 | | 6,671 ± 121.4 | | 0.257 ± 0.002 | | 0.004 ± 0.001 |
| ERGM (27% EO) | **16,346** | ± 101 | 0.945 ± 0.001 | | 1,753 ± 15 | | *80,510* ± 1,337 | | *0.474* ± 0.003 | | 0.27 | ± 0.01 |
| CORA-ML | 101,872 | | 0.941 | | 2,810 | | 3.1e6 | | 0.482 | | 1 | |
| GraphGAN (39% EO) | 75,724 | ± 1,401 | 0.959 ± 0.000 | | **2,809** ± 1.6 | | 1.8e6 ± 141,795 | | 0.398 ± 0.002 | | 0.39 | ± 0.004 |
| GraphGAN (52% EO) | *86,763* | ± 1,096 | 0.954 ± 0.001 | | 2,807 ± 1.6 | | *2.6e6* ± 103,667 | | *0.42* ± 0.003 | | 0.52 | ± 0.001 |
| DC-SBM (11% EO) | 73,921 | ± 3,436 | 0.934 ± 0.001 | | 2,474 ± 18.9 | | 1.2e6 ± 170,045 | | 0.523 ± 0.003 | | 0.11 | ± 0.003 |
| Conf. model | * | * | 0.928 ± 0.002 | | 2,785 ± 4.9 | | * | * | * | * | 0.013 ± 0.001 |
| Conf. model (39% EO) | * | * | 0.931 ± 0.001 | | 2,793 ± 2.0 | | * | * | * | * | 0.39 | ± 0.0 |
| Conf. model (52% EO) | * | * | *0.933* ± 0.001 | | 2,793 ± 6.0 | | * | * | * | * | 0.52 | ± 0.0 |
| node2vec naïve | 31,456 | ± 91.5 | 0.990 ± 0.000 | | 2,810 ± 0.1 | | 47,548 ± 516 | | 0.226 ± 0.002 | | 0.006 ± 0.001 |
| ERGM (56% EO) | **98,615** | ± 385 | 0.932 ± 0.001 | | 2,489 ± 11 | | **3,1e6** ± 57,092 | | **0.517** ± 0.002 | | 0.56 | ± 0.014 |

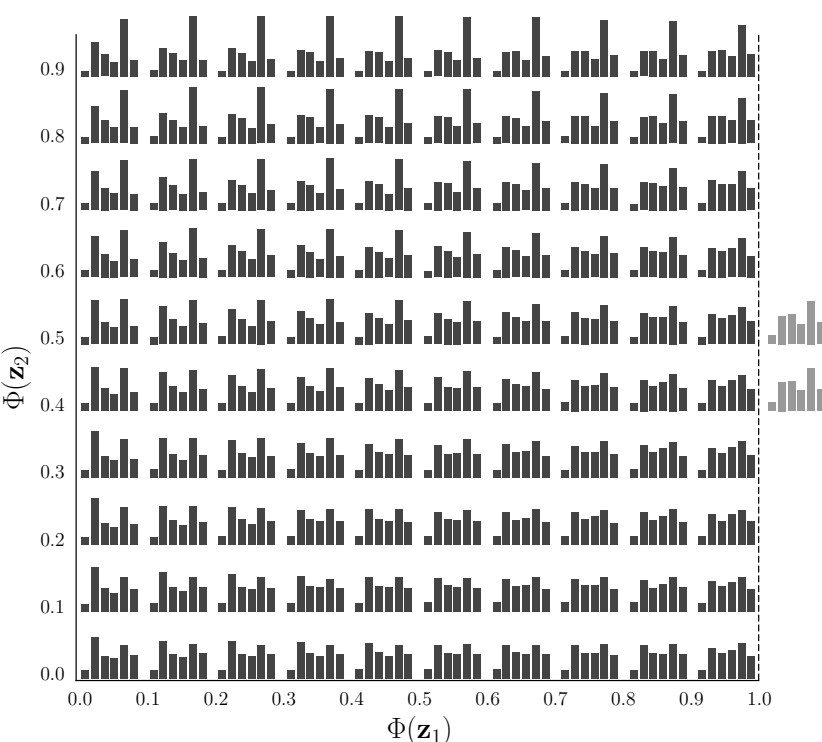

Figure 12: Community distributions of graphs generated by GraphGAN on subregions of the latent space $z$, trained on the CITESEER network.

