# OpenReview forum: "GraphGAN: Generating Graphs via Random Walks"
_ICLR.cc/2018/Conference — Reject_

### Official Review · AnonReviewer2 · 2017-11-27
**Positive about manuscript but it needs improvement**

**Rating:** 6
**Confidence:** 4

**Review:**

I am overall positive about the work but I would like to see some questions addressed.

Quality: The paper is good but does not address some important issues. The paper proposes a GAN model to generate graphs with non-trivial properties. This is possibly one of the best papers on graph generation using GANs currently in the literature. However, there are a number of statistical issues that should be addressed. I fear the paper is not ready yet, but I am not opposed to publication as long as there are warnings in the paper about the shortcomings.

Originality: This is an original approach. Random walks sometimes are overused in the graph literature, but they seem justified in this work. But it also requires extra work to ensure they are generating meaningful graphs.

Significance: The problem is important. Learn to generate graphs is a key task in drug discovery, relational learning, and knowledge discovery.

Evaluation: The link prediction task is too easy, as links are missing at random. It would be more useful to predict links that are removed with an unknown bias. The graph (wedge, claw, etc) characteristics are good (but simple) metrics; however, it is unclear how a random graph with the same size and degree distribution (configuration model) would generate for the same metrics (it is not shown for comparison).

Issues that I wish were addressed in the paper:
a)	How is the method learning a generator from a single graph? What are the conditions under which the method is likely to perform well? It seems to rely on some mixing RW conditions to model the distinct graph communities. What are these mixing conditions? These are important questions that should have at least an empirical exploration.
b)	What is the spatial independence assumption needed for such a generator?
c)	Would this approach be able to generate a lattice? Would it be able to generate an expander graph? What about a graph with poorly connect communities? Is there any difficulties with power law graphs?
d)	How is the RW statistically addressing the generation of high-order (subgraph) features?
e)	Can this approach be used with multiple i.i.d. graphs?
f)	Isn’t learning the random walk sample path a much harder / higher-dimensional task than it is necessary? Again, the short walk may be capturing the communities but the high-dimensional random walk sample path seems like a high price to pay to learn community structure.
g)	Clearly, with a large T (number of RW steps), the RW is not modeling just a single community. Is there a way to choose T? How larger values of T to better model inter-community links? Would different communities have different choices of T?
h)	And a related question, how well can the method generate the inter-community links?
i)	The RW model is actually similar to an HMM. Would learning a mixture of HMMs (one per community) have similar performance?

---

> ### Author Response · Authors · 2017-12-08
> **Authors' answer pt. 1**
>
> Thank you for your review and constructive feedback.
>
> We noticed that several of your questions (a, c, f, g, h, i) revolve around community structure. We would like to highlight that our model does not have access to community information at any point and more importantly does not have the goal of explicitly modeling communities. We address all your concerns below and additionally as you requested, we extended the discussion section to clearly highlight the model limitations (see Section 5 in the revised manuscript).
>
> 1) Link prediction
> We replicated the experimental setup for link prediction that is standard in recent works ([1, 2, 3]). We agree that different test set sampling strategies could provide a more in-depth analysis of our link prediction performance. However, our main goal is to demonstrate the feasibility and utility of implicit generative modeling for graphs, and not to develop a new state-of-the-art method for link prediction. This experiment mainly serves the purpose of demonstrating the generalization properties of the proposed method.
>
> On a related note, since implicit models for graph generation have not been studied so far, effective methods for their evaluation are yet to be developed. Therefore, we use the link prediction task as one possible way to evaluate our implicit model. This, together with the other experiments, gives us insight into the graph properties that our model is able to capture.
>
> 2) Configuration model
> Thank you for this suggestion. We have added results for the configuration model to Table 2 in the revised version, as well as to Table 6 in the appendix. Some properties of the graphs generated by the configuration model (e.g., degree distribution) are identical to the input graph statistics by the definition of the configuration model. However, the random edge shuffling performed by the configuration model completely destroys the community structure, which makes the resulting graph very different from the original.
>
> Additionally, we have performed experiments, where only a fraction of edges are rewired by the configuration model, such that the edge overlap (EO) score of the rewired graph matches the EO score of GraphGAN. Still, even in such a scenario, the configuration model significantly alters the community structure. You can see the quantitative results in Table 2 of the revised version of the paper.
>
> Regarding your questions (a) - (c) (questions (d)-(i) are in pt. 2)
> a)
> Indeed, learning graph generators from a single graph is one of the key challenges tackled in our paper. In fact, part of our motivation for using RWs was precisely to solve this challenge (see also (f)). Given the nature of the GAN framework we required multiple samples to train the generator. Thus we turn to using RWs since they naturally represent our single input graph with multiple samples.
>
> In this first foundational work we explored connected graphs  (by extracting the largest connected component as a preprocessing step). We did not investigate the behaviour of GraphGAN when we have e.g. many disconnected components. This could be considered as one condition for GraphGAN to perform well.
>
> Furthermore, the focus of this paper was to show that implicit graph generators are able to capture properties of the graph without manually specifying them. While our goal is not to determine the stationary distribution of RWs (for which mixing conditions are relevant), we agree that drawing theoretical connections between GraphGAN and the established results is an exciting direction for future work. Please note that some empirical exploration of this aspect is already included (Figure 4, where we analyze the effect of RW length on link prediction performance). See also our answer to (g).
>
> b)
> Our model does not make *any* spatial dependence assumptions about the adjacency matrix, assuming you are referring to our discussion in paragraph 2 of Section 2. The main point in the paper is that one should not naively treat the adjacency matrix as a binary image and apply standard CNN-based GAN architectures to it. Such architectures for images contain the built-in assumption that pixels located closely (within the same receptive field) are in some way correlated. Clearly, when talking about an adjacency matrix such assumption is not sensible, as permutations of rows/columns correspond to the exact same graph but very different receptive fields. Our model addresses this issue by operating on the random walks.
>
> c)
> We are indeed able to generate graphs with power-law degree distributions and sparse connectivity. We can conclude this since our model was evaluated and shows good performance on real-world graphs, that all exhibit exactly those patterns (see Table 6). Since our focus is on complex real-world networks, we felt that experiments on toy graphs (lattice, expander) would distract from the main story.

---

> > ### Author Response · Authors · 2017-12-08
> > **Authors' answer pt. 2**
> >
> > d)
> > Given our architecture we can capture high-order (subgraph) features, despite not explicitly modeling them. This is due to two different reasons, both involving memory.
> >
> > First, since our generator is based on an LSTM unit -- which has memory -- it can capture high-order interactions and utilize them during generation. Note that when generating the n-th node in a RW the LSTM has access to the entire history of nodes generated before and can utilize this history/memory to encode high-order features.
> >
> > Secondly, the input we feed to GraphGAN are second-order random walks as introduced in node2vec [2]. The memory factor records each previous step and influences the walking direction, leading to a biased random walk, essentially having a trade-off between breadth-first search (BFS) and depth-first search (DFS).
> >
> > e)
> > This depends on what exactly is meant by the i.i.d. property. In case all graphs have the same number / ordering of nodes, it is straightforward to apply GraphGAN to them. However, in this work we wanted to focus on the single graph setting -- which is common in many fields.
> >
> > We agree that applying our model to collections of smaller i.i.d. graphs (such as chemical compounds or molecules) is an interesting and important research question. However, this setting will require very different evaluation protocols, (and potentially, appropriate alterations in the model architecture), which is why we leave it to follow-up work.
> >
> > f)
> > Learning to generate RWs is a conscious decision, motivated by our intention to solve two key challenges: permutation invariance and learning from a single graph. First, by learning to generate RWs (compared to say learning to generate the full adjacency matrix) our model is invariant to arbitrary permutations of the nodes. Second, the class of implicit models we are considering requires multiple samples for training. Thus, we turn to using RWs which naturally represent the input graph. In short, by using RWs we were able to solve 2 out of 3 key challenges in learning implicit graph models and they are thus crucial to the success of our model. This also relates to question (d) where we talk about why the RWs are able to capture such properties.
> >
> > Regarding the “high price”: We want to highlight (see also answers to (g), (h), (i)) that communities are *not* explicitly modeled by GraphGAN (nor they are available during training) and our goal is not to learn the community structure. The focus is on showing that implicit graph generation models are able to capture important graph properties (whatever those might be, possibly including communities) without manually specifying them beforehand.
> >
> > g & h)
> > Please note that we have already performed an experiment that shows how we can choose the number of RW steps T. Figure 4 shows the link prediction performance as the length of the random walks increases.  We observe that while the performance for RWs of small length (T <=4) is not satisfactory, having RWs of length 16 already yields competitive link prediction performance. Furthermore, we observe that the performance for RW length 20 over 16 is marginal and does not outweigh the additional computational cost (note that the number of model parameters does not increase with T because of the recurrent architecture). In short, we can choose T empirically by looking that link prediction performance.
> >
> > As described in Section 4.2, in our experimental setup for link prediction we hold out 10% / 5 % of edges for the validation / test set respectively, along with the same number of non-edges. Since these edges/non-edges are selected completely at random, the validation/test set contains both inter- and intra-community links. This coupled with the fact that our link prediction performance is competitive (see Table 3) clearly shows that our method is able to model/generate the inter-community edges very well.
> >
> > Again, please note that communities are not explicitly modeled or even available to GraphGAN during training. The focus is on implicit graph generation. The implicit model does end up learning about the community structure, since it turns out to be useful for graph generation, but this was not manually incorporated/modeled from our side.
> >
> > i)
> > As previously mentioned our model is learned in a completely unsupervised fashion, i.e., the community information is not available to GraphGAN during training. More importantly, we do not even want to explicitly model the communities, since the main goal of our work was to learn an implicit graph generator. Your suggestion relies on apriori community information, which is not available in our case.
> >
> > References
> > ---------------
> > [1] Grover, Aditya, and Jure Leskovec. "node2vec: Scalable feature learning for networks." KDD’16.
> > [2] Wang, Daixin, Peng Cui, and Wenwu Zhu. "Structural deep network embedding." KDD’16.
> > [3] Kipf, Thomas N., and Max Welling. "Variational Graph Auto-Encoders." arXiv:1611.07308 (2016).

---

### Official Review · AnonReviewer3 · 2017-11-27
**A pioneering work with great potential**

**Rating:** 7
**Confidence:** 4

**Review:**

The authors proposed a generative model of random walks on graphs. Using GAN, the architecture allows for model-agnostic learning, controllable fitting, ensemble graph generation. It also produces meaningful node embeddings with semi-interpretable latent spaces. The overall framework could be relevant to multiple areas in graph analytics, including graph comparison, graph sampling, graph embedding and relational feature selection. The draft is well written with convincing experiments. I support the acceptances of this paper.

I do have a few questions that might help further improve the draft. More baseline besides DC-SBM could better illustrate the power of GAN in learning longer random walk trajectories. DC-SBM, while a generative model, inherently can only capture first order random walks with target degree biases, and generally over-fits into degree sequences. Are there existing generative models based on walk paths?

The choice of early stopping is a very interesting problem especially for the EO-creitenrion. In Fig3 (b), it seems assortativity is over-fitted beyond 40k iterations. It might be helpful to discuss more about the over-fitting of different graph properties.

The node classification experiment could use a bit more refinement. The curves in Fig. 5(a) are not well explained. What is the "combined"? The claim of competitive performance needs better justification according to the presentation of the F1 scores.

The Latent variable interpolation experiment could also use more explanations. How is the 2d subspace chosen? What is the intuition behind the random walks and graphs of Fig 6? Can you provide visualizations of the communities of the interpolated graphs in Fig 7?

---

> ### Author Response · Authors · 2017-12-08
> **Authors' answer**
>
> Thank you for comments. Based on your comments, we have a uploaded a revised version of our paper. See the details below.
>
> 1) Baselines
> Since our main goal is to develop an implicit model for graph generation the focus of the experimental evaluation was to show that the implicitly generated graphs are useful, rather than outperforming existing explicit generators (most of which are designed with specific graph patterns in mind). Thus, we chose the well established DC-SBM as a baseline. In the revised version we have added new baselines (on the suggestions of the reviewers) such as the configuration model and ERGM. You can find the results for these in Table 2. We are open to suggestions about further graph generation models we could compare to in order to highlight the properties and limitations of GraphGAN.
>
> 2) Generative models based on RWs
> The only remotely related model we are aware of that involves random walks is the Butterfly model (McGlohon et al., “Weighted graphs and disconnected components: patterns and a generator", KDD’08). However, it is a variant of the preferential attachment principle, and focuses on networks that grow over time. It uses first-order random walks, and thus cannot capture higher-order interactions. Moreover, it is not applicable to our task since it generates the random walks on the fly (i.e., is not based on a set of random walks as input).
>
> 3) EO-criterion
> This is an interesting point, and we briefly talk about it in Section 3.2. We can view the VAL and EO criteria as a trade-off between better generalization (in terms of link prediction) and more accurate reconstruction of the graph. Thus, as the EO score increases, our model approximates the distribution of random walks in the original graph more closely, which leads to constructing graphs more similar to the input, up to the point of overfitting the input graph. The EO criterion gives us control over this.
>
> 4) Node "embeddings" (Figure 5)
> Thank you for pointing this out. We have updated this section and will briefly summarize the changes in the following. Using the term ‘embedding’ was unfortunate in the figure. We were referring to W_down, i.e. the weight matrix projecting down the one-hot node vectors into low-dimensional space. The term ‘context’, on the other hand, was referring to W_up, which projects up the low-dimensional LSTM output. ‘Combined’ was the concatenation of the W_up and W_down, which gave an additional small improvement over only using W_up. When using the term ‘competitive’ we were referring to the link prediction performance, and not node classification; we have made this distinction more clear in the revised version (see, e.g., the updated abstract).
>
> 5) Latent space interpolation (Figures 6, 7)
> We improved the wording in the revised paper. To summarize: The generator takes as input a noise vector z, drawn from a d-dimensional standard normal distribution. For the latent space interpolation experiment we set d=2. That is, the generator receives samples drawn from a bivariate Gaussian distribution and transforms these into random walks. By using the inverse of the cumulative distribution of this 2D Gaussian distribution we can divide the input domain (i.e. R^2) into bins of equal probability mass, and group the generated random walks into their respective bins based on the noise samples that were used to generate them. Based on these random walks, we construct the score matrix and assemble a graph using the procedure described in Section 3.3. We can now measure the properties of the random walks coming from each of the latent space bins (e.g. the average degree of the first node in the random walks: Figure 6a) as well as the graphs constructed from the random walks (e.g. Gini coefficient: Figure 6c) and observe how these properties smoothly change when interpolating in the latent space.
>
> To better visualize the latent space interpolation performed in Figure 7, we have compiled a short animation (https://figshare.com/articles/GraphGAN_Latent_Space_Interpolation/5684137). In the bottom-right section, you can observe how the generated graphs’ structure changes when interpolating along a trajectory in the latent space.

---

### Official Review · AnonReviewer1 · 2017-12-02
**Claims and evaluation need some work**

**Rating:** 4
**Confidence:** 5

**Review:**

This paper proposes a WGAN formulation for generating graphs based on random walks. The proposed generator model combines node embeddings, with an LSTM architecture for modeling the sequence of nodes visited in a random walk; the discriminator distinguishes real from fake walks.

The model is learned from a single large input graph (for three real-world networks) and evaluated against one baseline generative graph model: degree-corrected stochastic block models.

The primary claims of the paper are as follows:
i) The proposed approach is a generative model of graphs, specifically producing "sibling" graphs
ii) The learned latent representation provides an interpretation of generated graph properties
iii) The model generalizes well in terms of link and node classification

The proposed method is novel and the incorporated ideas are quite interesting (e.g., discriminating real from fake random walks, generating random walks from node embeddings and LSTMs). However, from a graph generation perspective, the problem formulation and evaluation do not sufficiently demonstrate the utility of proposed method.

First, wrt claim (i) the problem of generating "sibling" graphs is ill-posed. Statistical graph models are typically designed to generate a probability distribution over all graphs with N nodes and, as such, are evaluated based on how well they model that distribution. The notion of a "sibling" graph used in this paper is not clearly defined, but it seems to only be useful if the sibling graphs are likely under the distribution. Unfortunately, the likelihood of the sampled graphs is not explicitly evaluated. On the other hand, since many of the edges are shared the "siblings" may be nearly isomorphic to the input graph, which is not useful from a graph modeling perspective.

For claim (i), the comparison to related work is far from sufficient to demonstrate its utility as a graph generation model. There are many graph models that are superior to DC-SBM, including KPGMs, BETR, ERGMs, hierarchical random graph models and latent space models. Moreover, a very simple baseline to assess the LSTM component of the model, would be to produce a graph by sampling links repeatedly from the latent space of node embeddings.

Next, the evaluation wrt to claim (ii) is novel and may help developers understand the model characteristics. However, since the properties are measured based on a set of random walks it is still difficult to interpret the impact on the generated graphs (since an arbitrary node in the final graph will have some structure determined from each of the regions). Do the various regions generate different parts of the final graph structure (i.e., focusing on only a subset of the nodes)?

Lastly, the authors evaluate the learned model on link and node prediction tasks and state that the model's so-so performance supports the claim that the model can generalize. This is the weakest claim of the paper. The learned node embeddings appear to do significantly worse than node2vec, and the full model is worse than DC-SBM. Given that the proposed model is transductive (when there is significant edge overlap) it should do far better than DC-SBM which is inductive.

Overall, while the paper includes a wide range of experimental evaluation, they are aimed too broadly (and the results are too weak) to support any specific claim of the work. If the goal is to generate transductively (with many similar edges), then it would be better to compare more extensively to alternative node embedding and matrix factorization approaches, and assess the utility of the various modeling choices (e.g., LSTM, in/out embedding). If the goal is to generate inductively, over the full distribution of graphs, then it would be better to (i) assess whether the sampled graphs are isomorphic, and (ii) compare more extensively to alternative graph models (many of which have been published since 2010).

---

> ### Author Response · Authors · 2017-12-08
> **Authors' answer pt. 1**
>
> Thank you for your review.
>
> We would like to clarify some important points that might have been a source of misunderstanding. As this is the first work of its kind (implicit generative model for graphs), we neither expect nor claim that GraphGAN in its current form is superior to every existing explicit model in every possible regard. Rather, our goal is to lay a foundation for the study of implicit models for graph generation. Such models will let us capture important properties of real-world graphs, without having to manually specify them in our models.
>
> We are convinced that the results already confirm this statement, and show the feasibility and utility of implicit models. Still, based on your suggestions, we added a comparison with more baselines. As expected, and reinforcing our previous point, all properties which these approaches explicitly model are preserved, while the rest deviate significantly from the input graph. This highlights the need for implicit models, such as the one proposed in our paper. Furthermore, there exist properties not captured by any of the existing models as shown in [1], which again emphasizes the need for implicit models.
>
> We have already analyzed the reconstructive (graph statistics) and generalization (link prediction) properties of our model. As you mentioned, because of the likelihood-free nature of the model, we cannot evaluate the likelihood of the held-out edges or an entire sampled graph. We are not aware of other experimental protocols applicable to this novel problem setting. If you think that some important aspects are not evaluated, please let us know.
>
> Using your terminology GraphGAN would be considered inductive. In the revised version we include comparison with more baselines. However, we do not completely agree that “high edge overlap” => “transductive” (above which EO threshold does a model qualify?). This definition would mean that ERGM, which has high edge overlap (see Table 2), should be considered transductive, which it isn’t.
>
> We would also like to clarify, that our model is *not* generating random walks from node embeddings, and is not yet another method for learning node embeddings. Please, see our discussion on embeddings below.
>
> In the following comment we address your other concerns.

---

> > ### Author Response · Authors · 2017-12-08
> > **Authors' answer pt. 2**
> >
> > 1) Generalization
> > The problem of detecting (near-)isomorphism between two graphs is extremely challenging in general (when the nodes may be permuted). In our case, since the ordering in both the original and sibling graphs is identical, having low edge overlap directly implies that they are not (nearly) isomorphic, (note that the model is still invariant to node permutations). Additionally, given the strong link prediction performance, we can surely claim that the model does not simply "memorize" the original graph, and that the "sibling" graphs contain edges that are plausible but not present in the input graph.
> >
> > 2) Link prediction
> > Given the results in Table 3, the claim that our model achieves “so-so” link prediction is unjustified. Despite not being designed specifically for this task, we still outperform the competing methods in 4 out of 6 cases on smaller graphs. The less dominant performance on large graphs (>15k nodes) is clearly indicated in the paper, and the possible causes and solutions are mentioned in Section 5.
> >
> > 3) Node "embeddings"
> > As mentioned above, GraphGAN is neither designed to learn embeddings, nor is using established embedding approaches during the graph generation process. Using the term “embedding” was unfortunate and might have been a source of confusion. The so-called “embeddings” W_down & W_up are projection matrices between the low-dimensional LSTM space and the high-dimensional node space. Due to the lack of established evaluation techniques for implicit generative graph models, we decided to discuss the properties of the these matrices to give the reader a better insight about the behavior of the model. We made this point more clear in the revised version of the paper.
> >
> > 4) Baselines
> > As per your suggestion, we now additionally compare against further baselines:
> >
> > 4.1) Sampling graphs from node embeddings
> > On your suggestion, we repeatedly sampled links from the latent space of node embeddings, and included the results in Table 2 the revised version (“naive node2vec”). As we can see, such procedure leads to a dramatically worse reconstruction, which justifies the use of an LSTM. Poor performance of embedding-based graph generation makes sense, as that is not what the embeddings are designed to do.
> >
> > 4.2) ERGM
> > While ERGMs [2] do well at reconstructing the graph statistics that are explicitly modeled by the s(G) term (such as degree distribution, assortativity, average edge density), they perform significantly worse when it comes to other metrics (e.g., community structure, LCC size), as can be seen in Table 6.
> >
> > This fundamental limitation is exactly the reason we turn to implicit models in the first place: We want to have a model that automatically detects patterns in graph structure and generates new networks that follow them, without having to manually specify them.
> >
> > 4.3) Configuration model
> > Based on another reviewer’s comment we have added a comparison with the configuration model. Similarly to ERGM, it preserves only those characteristics, for which it was explicitly designed for.
> >
> > 5) Latent space interpolation
> > One of your concerns was that the impact of the latent space on the generated graphs is not clear. In fact, Figure 6c&d, Figure 7 and Figures 10&11 in the appendix (namely subfigures, c, d, e, f, g, h, i, j, o, p) specifically measure properties of the entire graphs, not the random walks.
> >
> > Furthermore, the various regions of the latent space are clearly responsible for generating graphs with noticeably different structure as you can see in Figure 7 as well as in this animation that interpolates in the latent space (https://figshare.com/articles/GraphGAN_Latent_Space_Interpolation/5684137).
> >
> > References
> > [1] Yuxiao Dong et al., “Structural diversity and homophily: A study across more than one hundred big networks”, KDD’17.
> > [2] Hunter, David R., et al. "ergm: A package to fit, simulate and diagnose exponential-family models for networks." Journal of statistical software 24.3 (2008): nihpa54860.
> > [3] Hamilton, William L., Rex Ying, and Jure Leskovec. "Inductive Representation Learning on Large Graphs." arXiv preprint arXiv:1706.02216 (2017).

---

### Public Comment · ~Jiaxuan_You1 · 2017-11-06
**About latent space interpolation experiment**

It's a great paper to read. I have a question on the latent space interpolation experiment of the paper.
If I got it right, you do random sampling in hidden space to produce the results in Table 2, and the statistics seem to be pretty stable. However, when you do latent space interpolation, the statistics seem to vary a lot. Why does this happen?
You mention that "certain regions of z correspond to generated graphs with very different degree distributions", however it Figure 3(a), by random generating a graph, the degree distribution matches the ground truth well. I'm confused about why the latent space has such kind of property.
Maybe I made some mistakes when trying to understand the experiment. Looking forward to your reply! Thanks!

---

> ### Author Response · Authors · 2017-11-09
> **Re: About latent space interpolation experiment**
>
> Thank you very much for your interest in our paper and your comment.
>
> It seems that the source of confusion is that on the one hand, we show that graphs generated using random sampling from the latent space are similar to the input graph (Table 2, Fig. 3a), while on the other hand, in the latent space interpolation (Fig. 6 and 7), the generated graphs have very different properties compared to the input graph.
>
> Let's recap how the latent space interpolation is performed for clarity. Remember that a single noise vector does not produce a complete graph, but rather one random walk. We therefore sample a large number of random walks from the latent space and use the method described in Sec. 3.3 to assemble a graph from these random walks.
>
> If we now restrict the sampling to specific subregions of the latent space, intuitively, we obtain random walks that have some specific properties, which in turn makes the graphs assembled from them have specific properties. However, if we sample from the entire latent space, we are in a way "averaging" over all of these properties and the sampled random walks (and the resulting graph) have similar properties as the original, e.g. as you have noticed in Table 2 and Figure 3a.
>
> We hope that this answer helps you to better understand our experiment. Please do not hesitate to comment again if you have any other or follow-up questions.
>
> tl;dr: Specific regions of the latent space encode specific properties, the "average" over all regions has properties similar to the original.

---

### Public Comment · (anonymous) · 2017-11-24
**relation with a recent paper**

An AAAI18 paper released recently also propose a graph GAN framework https://arxiv.org/abs/1711.08267, what's the difference between this paper and their paper? It seems that their results is more dominant in link prediction than this paper.

---

> ### Author Response · Authors · 2017-11-24
> **Re: relation with a recent paper**
>
> While both models coincidentally have the same acronym and use the GAN framework, they are very distinct in their nature and have different goals.
>
> The model proposed in the paper you referenced is an explicit (prescribed) probabilistic model whose goal is to learn node embeddings. The explicitly specified probability distribution G(v | v_c) can be computed directly given the embedding \theta_G (Equation 5). Such model could also be learned by other means (e.g., by directly minimizing cross entropy + negative sampling for non-edges), and the use of GANs in this setting is rather unconventional.
>
> In contrast, our approach defines an implicit generative model for random walks in the graph. Its main goal is to generate new graphs that have similar properties to original (but are not exact replicas). As is the case for implicit models, samples can be drawn from it, but direct computation of the probabilities is not possible. In such a scenario, GAN training is one of the few available options. Our implicit model is not restricted to pairwise interactions and can capture higher-order properties of the graph.
>
> Note, that link prediction is the optimization objective in the work you mentioned. Thus, it is not surprising that the obtained node embeddings achieve high scores in the related tasks. Meanwhile, our model is not trained for link prediction, and the embeddings are just a byproduct of the learning process.
>
> In addition to pointing out these fundamental differences, we would also like to highlight that the above-mentioned work was just made public on arXiv two days ago (Nov 22nd); which is why it could not be included in the Related Work section of our paper at the time of submission almost a month ago.
>
> TL;DR: While at first glance the approaches appear to be related (both are called GraphGAN), after carefully reading the papers, it becomes clear that the two models are fundamentally different and have orthogonal goals. The other work: explicit model + pairwise interactions for learning node embeddings. Our work: implicit model + higher-order interactions, with the goal of generating new graphs.

---

### Public Comment · ~Junliang_Guo1 · 2017-11-27
**The constructed matrix S while training with EO early stop strategy**

It's a very interesting work!  There are two parts that I'm confused after reading the paper:

1. In Section 3.2, while training with EO-Criterion early stopping strategy, you construct a score matrix S at every validation step. How is S constructed in EO? In Val-Criterion, S is constructed through 1k recently generated random walks. While in EO, is S still constructed the same as in Val, or generated through 500k generated random walks as you described in Section 3.3? It's time-consuming if you construct S from a large corpus of random walks at every validation iteration, and if you use the small corpus as used in Val, how to guarantee the approximation error of edge overlap ratio is bounded, i.e., won't be too large to damage the performance?

2. This work generates sibling graphs of the original graph. In what applications can we utilize this method? Normal graph generative models generate a relevant node given a prior node, but this paper generates a new graph given a prior graph, thus seems cannot be directly used in node-level graph applications. Any references will be better :)

Looking forward to your reply! Thanks!

---

> ### Author Response · Authors · 2017-11-27
> **Re: The constructed matrix S while training with EO early stop strategy**
>
> Thank you for your comment and interest in our work!
>
> 1) For the EO criterion, S is constructed the same as in Val. In both cases S is constructed based on the last 1k iterations (not 1k random walks), which typically amounts to around 750k random walks (depending on the specific setting). Thus, we expect the approximation to be reasonably good. Note, that we update the list of the RWs generated in the last 1k iterations incrementally (in a sliding window / queue fashion). This means that at every iteration we only need to subtract the 750 oldest entries from S, and add the 750 newest, which is highly efficient.
>
> 2) We are not sure whether we correctly understood your definition of a generative model. We base our notion of a generative model for graphs on [1]. In this context, such a model is used to generate entire graphs that exhibit desired properties (e.g., matching the properties of a given input graph). Some examples include the Configuration Model [2] and Barabási–Albert Model [3]. While the purpose of GraphGAN is to generate entire graphs, we can also use it for node-level tasks such as link prediction, as shown in the experimental section.
>
> As for the concrete application scenarios, we again refer to [1]. One case where our model is readily applicable is simulation studies (using the language of [1]). Imagine that we are developing a new algorithm for some graph-related problem, e.g. community detection. Often, we don't have access to much labeled data that all comes from the same distribution. However, GraphGAN still lets us estimate how our new algorithm will behave in the wild. For this, we can create sibling graphs using GraphGAN and evaluate performance of the new algorithm on them.
>
> There are surely many other tasks that GraphGAN could be applied to, that we leave for follow-up work, such as anomaly detection, graph compression, data anonymization, etc.
>
> We hope this answer clarifies the uncertainties you had about our work. Please do not hesitate to post follow-up questions.
>
> References:
> [1] Deepayan Chakrabarti and Christos Faloutsos. Graph mining:  Laws, generators, and algorithms.
> ACM computing surveys (CSUR), 38(1):2, 2006.
> [2] http://homepage.divms.uiowa.edu/~sriram/196/spring12/lectureNotes/Lecture11.pdf
> [3] Albert-Laszlo Barabasi and Reka Albert.  Emergence of scaling in random networks. Science, 286 (5439):509–512, 1999.

---

> > ### Public Comment · ~Junliang_Guo1 · 2017-11-28
> > **one more question**
> >
> > Thanks for your clear reply! And one more question:
> >
> > In Section 3.1, the next sample is generated as v_{t} = onehot(argmax v_{t}^{*}). How is this step differentiable? As argmax is a hard assignment, the gradients cannot be passed to v_{t}^{*} during backward as you claimed. Maybe I misunderstand somewhere?

---

> > > ### Author Response · Authors · 2017-11-28
> > > **Re: one more question**
> > >
> > > We use the Straight-Through Gumbel-Softmax estimator that is described in [1]. In a nutshell, this allows us to approximate sampling from a categorical distribution in a differentiable way.
> > >
> > > [1] Jang, Eric, Shixiang Gu, and Ben Poole. "Categorical reparameterization with Gumbel-softmax." ICLR 2017

---

### Public Comment · (anonymous) · 2017-12-03
**Issue about duplicate model name**

Neat Work,
But I found that there was another paper named "GraphGAN" on ArXiv: https://arxiv.org/abs/1711.08267, which has been accepted by AAAI 2018.
It might be confusing for readers to distinguish these two models.

---

> ### Author Response · Authors · 2017-12-04
> **Re: Issue about duplicate model name**
>
> As stated in our reply to an earlier comment (see below), we are aware of this; while we believe that both works are very distinct, we are considering alternative names to avoid confusion.

---

### Author Response · Authors · 2018-01-05
**Revision summary**

Based on the reviewers' comments we have made the following improvements to our paper:
* Added more details on the experimental setup (Section 4.4).
* Clarified the role of the embedding-like matrices W_up and W_down (Section 4.3).
* Added comparisons with more baselines (Sections 2 & 4.1).
* Extended the discussion of the model's limitations & future work (Section 5).
* Fixed several typos and improved wording in a few places.

On the request of the program chairs, we would like to provide pointers to related papers that are also under submission to ICLR2018. While multiple deep generative models for graphs are proposed (e.g., https://openreview.net/forum?id=Hy1d-ebAb, https://openreview.net/forum?id=SJlhPMWAW, https://openreview.net/forum?id=BJcAWaeCW), our work is the only one that focuses on the single large real-world graph setting.

---

### Decision · Program_Chairs · 2018-01-29
**ICLR 2018 Conference Acceptance Decision**

**Decision:**

Reject

**Comment:**

This paper proposes an implicit model of graphs, trained adversarially using the Gumbel-softmax trick.  The main idea of feeding random walks to the discriminator is interesting and novel.  However,
1) The task of generating 'sibling graphs', for some sort of bootstrap analysis, isn't well-motivated.
2) The method is complicated and presumably hard to tune, with two separate early-stopping thresholds that need to be tuned
3) There is not even a mention of a large existing literature on generative models of graphs using variational autoencoders.